# Advances in Chip-Based Quantum Key Distribution

**DOI:** 10.3390/e24101334

**Published:** 2022-09-22

**Authors:** Qiang Liu, Yinming Huang, Yongqiang Du, Zhengeng Zhao, Minming Geng, Zhenrong Zhang, Kejin Wei

**Affiliations:** 1Guangxi Key Laboratory of Multimedia Communications and Network Technology, School of Computer, Electronics and Information, Guangxi University, Nanning 530004, China; 2Guangxi Key Laboratory for Relativistic Astrophysics, School of Physical Science and Technology, Guangxi University, Nanning 530004, China

**Keywords:** quantum key distribution, integration technologies, chip-based QKD

## Abstract

Quantum key distribution (QKD), guaranteed by the principles of quantum mechanics, is one of the most promising solutions for the future of secure communication. Integrated quantum photonics provides a stable, compact, and robust platform for the implementation of complex photonic circuits amenable to mass manufacture, and also allows for the generation, detection, and processing of quantum states of light at a growing system’s scale, functionality, and complexity. Integrated quantum photonics provides a compelling technology for the integration of QKD systems. In this review, we summarize the advances in integrated QKD systems, including integrated photon sources, detectors, and encoding and decoding components for QKD implements. Complete demonstrations of various QKD schemes based on integrated photonic chips are also discussed.

## 1. Introduction

### 1.1. Secure Communication

With the rapid development of communication technologies and computing science, communication systems and associated information-processing technologies have been widely used in engineering, commerce, and all aspects of human daily life. However, informatization brings not only convenience and efficiency, but also security risks. For instance, Microsoft Exchange Server vulnerabilities have been exploited, and resulted in tens of thousands of enterprises being attacked [1]. Cyberattacks in Iran led to the mass closures of gas stations across the country [2]. The supervisory control and data acquisition system of Florida drinking water-treatment facilities were invaded, and residents’ lives were threatened [3]. Cyberwarfare can undermine the security of critical infrastructures and have a significant impact on the global economy and human life. The security of cyberentities is considered as an important national strategic priority.

The confidentiality of information is an extremely important link of network security. The information cryptography used in modern communication systems based on the assumption of the unidirectional properties of some mathematical analyses, such as the decomposition of large numbers or the solution of discrete logarithms. In other words, the forward calculation (encryption process) of these mathematical problems is very simple, whereas the reverse calculation (decryption process) is extremely complex. Hence, the computation of solving eavesdropping-related problems will be far greater than that of solving encryption-related problems. Even if an eavesdropper’s computing capacity is much better than that of the encryptor, she cannot complete the decoding within the secrecy period of transmitted messages. However, with the development of computing science, the promotion of computing capability has far exceeded the initial imagination, so the security of current cryptography may no longer be reliable. In particular, quantum computing theoretically smashes the assumption of computational unidirectionality of mathematical analyses, which is the basic principle of traditional cryptography. Recently, small-scale quantum computers and quantum supremacy have been reported [4,5,6,7,8]. Although the realization of large-scale quantum computers may be decades away, its potential threat to current information security cannot be ignored. Interestingly, before people realized that quantum computers could be used to crack the current cryptographic systems, they had found approaches to deal with this threat.

Broadly speaking, there are three methods to the quantum secure encryption scheme. One method is to continue using the traditional public key cryptography but develop alternative algorithms to resist quantum attacks. This method is named postquantum cryptography (PQC) [9]. Its technological merit is that it can be compatible with existing cryptoinfrastructure, and it has usable high key rates over long distances. One shortcoming of PQC is that the developed algorithms have been proven to be safe only against known quantum attacks. This may result in future security vulnerabilities with potential disaster for information transmitted today.

The second method is quantum secure direct communication (QSDC), which was first proposed by Long and Liu in 2002 [10] based on quantum laws. QSDC allows users to directly transmit private information over secure quantum channels without security key distribution. Due to its relatively simple communication steps, QSDC has drawn great attention and has experienced rapid development [11,12,13,14] over the past two decades. However, the achieved key rates of QSDC [15,16,17,18] are rather lower than other quantum secure encryption schemes.

Another method is quantum key distribution (QKD) [19]. The unconditional security of QKD has been rigorously proven [20] based on quantum fundamental laws, such as quantum indistinguishability and the no-cloning theorem. Thus, its safety is independent of future improvements in computing capacity and algorithms. QKD combined with PQC is a typical quantum security scheme the safety of which can be theoretically proven at present. This scheme can instantly discover eavesdropping behaviors and events, and realize high-security and high-performance encryption systems.

### 1.2. Quantum Key Distribution

The QKD is a technique that allows two remote communication terminals to share a common secret key for cryptographic purposes. During the transmission between two parties (usually called Alice and Bob), an eavesdropper (usually called Eve) may eavesdrop on the quantum communication channel that spies on potential secret key bits. Eve unavoidably introduces disturbances to the transmitting quantum messages based on appropriate quantum laws, e.g., the quantum no-cloning theorem or Heisenberg uncertainty principle. Then the eavesdropping will be detected by the communicators and they can abandon such a key simply, then start new QKD process.

The first QKD scheme is named the BB84 protocol, which was originally proposed by Bennett and Brassard in 1979 and published in 1984 in a computer conference proceedings [19]. The well-known B92 protocol [21] is a version of the BB84 protocol revised in a cryptographic manner, and their physical natures are different. Both BB84 and B92 protocols are invented based on the indistinguishability properties of two arbitrary nonorthogonal qubits in a Hilbert space. In this paper, we refer the protocols, which are developed in a two-dimensional Hilbert space, as qubit-based QKD schemes. Other qubit-based QKD schemes include differential phase shift (DPS) [22] and coherent one way (COW) [23].

Although early QKD protocols use unentangled qubits as quantum information carriers, assorted protocols were proposed by using entanglement qubits [24] or qudits [25,26,27,28,29,30,31] instead. Qudit-based QKDs are developed in high-dimensional Hilbert spaces, and thus are also referred to as high-dimensional (HD) QKDs.

Generally, QKDs can be divided into two categories—discrete variable (DV) QKDs and continuous variable (CV) QKDs. We consider qubit-based and qudit-based QKDs as DV-QKDs. Unlike DV-QKD, the Hilbert space of the quantum state used for encryption in CV-QKD is infinite-dimensional and continuous.

Although QKD schemes own unconditional security in theory, there are still some security vulnerabilities in the practical QKD systems due to device imperfections. This issue was theoretically studied by Lütkenhaus [32] and Inamori et al. [33], and a notable framework for safety analysis of actual devices was established by Gottesman et al. [34]. Various protocols were also announced to deal with this issue. Relevant protocols include the decoy state [35,36,37], the Scarani–Acín–Ribordy–Gisin protocol [38], measurement device-independent (MDI) [39,40] and device-independent [41,42,43] protocols. For a more basic principle of QKD, we refer to two recent reviews: Xu et al. [44], who introduce the security of QKD with realistic device, and Pirandola et al. [45], who provide a general introduction and a comprehensive description of advances of QKD.

### 1.3. Focus and Outline of This Review

QKD has gradually matured after development over more than 30 years. The farthest distance of QKD via optical fiber link has exceeded 800 km [46]. Through quantum satellites, intercontinental QKD over one thousand kilometers has been achieved [47,48,49]. Commercial QKD systems have been already available on shelves. Some quantum networks have begun to build up, and relevant researches have been carried out [50,51].

However, there are still many challenges for large-scale QKD applications. In particular, there is a wide gap between existing QKD systems and traditional optical communication in terms of cost or integration level. Taking advantages of low cost, miniaturization, and industrial compatibility, integrated photonics show great potential for further promotion of QKD systems. In this review, we mainly concentrate on the advances in integrated chip-based photonic QKD.

The first step toward the application of integrated QKD is to realize functional QKD devices based on integrated platforms. The encoding and decoding of quantum states (normally photonic quantum states) are typical processes of QKD, and we summarize relative integrated devices in Table 1. In the past few decades, several integrated photon sources for QKD, including single-photon sources, weak coherent sources and entangled-photon sources, have been investigated, and their studies are listed in Table 2. Research of integrated quantum photonic detectors are collected in Table 3.

With the development of critical integrated devices, the principal verifications of QKD based on integrated platforms are gradually realized. Qubit-based QKD is the most mature QKD scheme, and related studies based on integrated systems are summarized in Table 4. We also pay attention to the applications of integration technology in some advanced QKD protocols, such as MDI-QKDs, which can be immune to all attacks on detectors. Relevant literature is summarized in Table 5. Furthermore, security analyses and CV-QKD and HD-QKD demonstrations based on integrated chips are collected in Table 6.

The outline of this review is as follows. In Section 2, we introduce the different integrated platforms for quantum photonics. In Section 3, the basic architecture of integrated QKD devices and implementations are discussed. In Section 4, we review the QKD demonstrations based on integrated platforms. In the last section, we provide some suggestions for future research.

Integration technology has an important impact not only on QKDs, but also on quantum computing and quantum information processing. Relevant studies include early reviews of advances in integrated quantum photonics [52,53] and silicon quantum photonics [54]. A list of reviews related to integrated quantum photonics is presented in Table 7.
entropy-24-01334-t001_Table 1Table 1QKD systems with integrated encoding and decoding modules.ReferencePlatformEncoding WayProtocolEncodingDecodingNotes[55]SiTime binBB84✔✔98% interference visibility[56]SiTime binBB84✔✔Interference visibility is 80% at 150 km[57]SiTime binBB84✔✔The lowest bit error rate is 0.7%[58]SiTime binBB84✔✔Silicon-based AMZI[59]SiTime binBB84✔✔Sifted key rate 2.4 kbps[60]SiTime binBB84✔✔45 km, WDM system[61]SiTime binBB84✔✔Stable WDM system for more than 30 days[62]SiPolarizationBB84✔✔Polarization extinction ratio greater than 20 dB[63]SiTime binBB84✔✔98.38% interference visibility[64]SiTime binBB84✔✔98.38% interference visibility[65]SiTime binBB84✔✔98.72% interference visibility[66]SiTime binBB84✔✔96% interference visibility[67]SiTime binBB84✘✔Non-blocking matrix switch[68]SiTime binDPS✘✔95.8% interference visibility[69]SiTime binBB84✘✔95.8% interference visibility[70]SiO_2_PolarizationBB84✘✔Polarization extinction ratio 16 dB[71]SiO_2_PolarizationBB84✘✔sifted key rate of 415 kbps[72]FLDWPolarizationMDI-QKD✘✔Bell state analyzer[73]SiTime binBB84✘✔Complementary decoding system[74]SiTime binBB84✘✔Low bit error rate AMZI[75]SiTime binBB84✘✔99% interference visibility[76]SiTime binBB84✘✔98.6% Interference visibility[77]LiNbO_3_ SiTime binBB84✘✔Extinction ratios are 18.65 dB and 15.46 dB[78]SiOAMHD-QKD✔✘Generating three OAM modes[79]SiPolarization-✔✘Polarization extinction ratio greater than 25 dB
entropy-24-01334-t002_Table 2Table 2QKD systems with integrated sources.ReferencePlatformTypeNotes[80]InPweak coherent-state source431 MHzHOM interference visibility is 46.5% ± 0.8%[81]InPcoherent-state source45 kHzside-mode suppression ratio of 54 dB[82]InPweak coherent-state source100 MHzHOM interference visibility is 46% ± 2%[83]Sientangled photon pair source431 MHzinterference visibility is 92%[84]LiNbO_3_ Sientangled photon sourceInterference visibility is 94%[85]Sientangled photon sourceHigh dimensional quantum information processing[86]Sientangled photon sourceQuantum information processing[87]Sientangled photon sourceQuantum information processing[88]hBNsingle-photon sourceIntegrated room temperature single-photon source
entropy-24-01334-t003_Table 3Table 3QKD systems with integrated detectors.ReferencePlatformEncoding WayProtocolSourceEncodingDecodingDetectionNotes[89]Si_3_N_4_Time binBB84✘✘✔✔BB84 system[90]SiTime binMDI-QKD✘✘✔✔MDI-QKD system
entropy-24-01334-t004_Table 4Table 4Qbit-based QKD of BB84, DPS and COW protocols based on integrated platforms.ReferencePlatformEncoding WayProtocolSourceEncodingDecodingDetectionClock Rate (Hz)Distance or LossKey Rate (kbit/s)[91]SiTime binBB84✘✔✔✘1.25 G45 km81.7[92]SiTime binBB84✘✔✔✘1.25 G14.5 dB200[93]SiPolarizationBB84✘✔✔✘10 M5 km0.95


DPS



1.72 G20 km565[94]InP SiO_*x*_N_*y*_Time binBB84✔✔✔✘560 M20 km345


COW



860 M20 km311[95]SiPolarizationTime binBB84✘✔✔✘1 G20 km329
Si SiO_*x*_N_*y*_Time binCOW



1.72 G20 km916[96]SiPolarizationBB84✘✔✘✘625 M43 km157[97]SiTime binBB84✘✔✔✘100 M20 km85.7[98]SiTime binBB84✔✔✘✘1 G100 km270


DPS



1 G100 km400[99]SiPolarizationBB84✘✘✔✘10 M20 km13.68[100]SiPolarizationBB84✘✔✘✘10 M145 m30


DPS



2 G14 dB400[101]SiO_*x*_N_*y*_Time binBB84✘✔✘✘2 G14 dB500


COW



2 G14 dB2500[89]Si_3_N_4_Time binBB84✘✘✔✔2.6 G2.5 dB1500[102]Si InPTime binBB84✔✔✔✘1 G25 km235[103]SiPolarizationBB84✘✔✔✘2 G20 km868[104]SiPolarizationBB84✘✔✘✘312.5 M100 km42.7[105]SiTime binBB84✘✔✔✘1.25 G50 km1340[106]SiTime binDPS✘✘✔✘1 G20 km3.076[107]SiTime binDPS✘✘✔✘1 G17.6 km120
entropy-24-01334-t005_Table 5Table 5MDI-QKDs based on integrated systems.ReferencePlatformEncoding WayProtocolSourceEncodingDecodingDetectionClock Rate (Hz)Distance or LossKey Rate (kbit/s)[108]SiPolarizationMDI-QKD✘✔✔✘0.5 M50 km1.46×10−3[109]InPTime binMDI-QKD✔✔✘✘250 M100 km1[110]SiPolarizationMDI-QKD✘✔✘✘1.25 G36 dB31×10−3[111]SiPolarizationMDI-QKD✘✔✘✘1.25 G24 dB137×10−3[90]SiTime binMDI-QKD✘✘✔✔125 M39.5 dB34×10−3
entropy-24-01334-t006_Table 6Table 6Other integrated QKDs.ReferencePlatformEncoding WayProtocolSourceEncodingDecodingDetectionClock Rate (Hz)Distance or LossKey Rate (kbit/s)Notes[112]SipathHD-QKD✘✔✔✘5 k4 dB7.5×10−3-[113]SiGaussian-modulatedCV-QKD✘✔✔✔250 M2 m250-[114]SiGaussian-modulatedCV-QKD✘✔✘✘---Security analysis[115]SiPolarizationMDI-QKD✘✔✘✘---Security analysis[116]SiPolarizationBB84✘✔✔✔---Security analysisentropy-24-01334-t007_Table 7Table 7Reviews of integrated quantum photonics.ReferenceNotes[117]Quantum communication[54]Silicon quantum photonics[118]Quantum photonic network[53]Photonic quantum information processing[119]Photonic quantum information processing[52]Photonic quantum information processing[120]Hybrid integrated quantum photonic circuits[121]Quantum entanglement on photonic chips[122]Femtosecond laser technology[123]Integrated photon technology[124]Silicon based quantum optical system devices[125]Direct phase modulated laser[126]Development, challenges, and directions of integrated quantum optics[127]Quantum communication and Quantum networks[128]Integrated photon-pair sources with nonlinear optics[129]Status, development and challenges of integrated quantum optics[130]Semiconductor quantum dot source and Quantum communication

## 2. Integrated Quantum Photonic Technology

The wide deployment of QKD requires a low-cost, robust, and miniature system. Integrated photonics provides potential for QKD devices in terms of complexity, robustness, and scalability. Many physical platforms have been studied for quantum implementations, and quantum photonic technology has raced ahead because of its relative simplicity in generation, manipulation, and detection. Serving as the carrier for quantum information, photons own the strength that can be inherently encoded in diverse degrees of freedom, involving temporal [131,132], frequency [133,134], path [85,135], and orbital angular momentum (OAM) [136,137]. In addition, multiple degrees of freedom of a photon can be used simultaneously [138]. To date, a series of photonic quantum implementations has been realized. As quantum systems scale up, the requirements for their stability, manufacturability, and programmability will be increasingly higher, and the miniaturization and chip-level integration of optical quantum implements are crucial to expand the complexity and functionality of quantum systems.

A range of photonic integration platforms has been investigated for quantum applications, including silica-on-insulator [139,140,141], silicon-on-insulator [142,143,144,145,146], silicon nitride (Si_3_N_4_) [147,148,149,150], silicon carbide [151], silicon oxynitride (SiO*_x_*N*_y_*) [95,133], lithium niobate (LN) [152,153,154], gallium arsenide (GaAs) [155,156,157], indium phosphide (InP) [95,158], and diamond [159], etc. Although the thickness of waveguides fabricated on silica-on-insulator is usually in the order of several microns whereas that of structures on silicon-on-insulator comes down to an order of hundreds of nanometers, both platforms can be referred to as planar lightwave circuits (PLCs) because the thickness of waveguides on a single chip is basically the same; that is, waveguides are in the same plane. Components created in the material platforms introduced above are typically 2D structures, whereas 3D photonic circuits can be integrated into a femtosecond laser direct written (FLDW) platform [160,161,162,163] because of the characteristics of fabrication processes. Different integrated platforms have their own superiorities. For instance, one prominent optical integration platform is nanostructures on silicon (Si) substrates. It has been used to realize various quantum facilities, including but not limited to the generation, detection, or manipulation of single-photon and high-dimensional entangled states [85], as well as some functions in chip-based quantum communication [94,110,113,164,165]. Another photonic integration standout is the FLDW platform. It has the typical advantage of arbitrary 3D circuit geometries in that it can be prepared by using the femtosecond laser direct writing technique. Many quantum tasks, such as 2D quantum walk [162] or homomorphic encryption [163], have been accomplished in this platform. The propagation loss of structures fabricated in Si_3_N_4_ platform is ultralow [150]. LN is an arising and flexible integrated platform for entangled photon sources [152,153], superconducting detectors [154], and high-speed modulators [166,167]. In addition to LN [153], GaAs [157] and InP [95] also exhibit obvious electro-optic characteristics, allowing fast processing of single photons.

## 3. QKD Implementation

A QKD implementation consists of three parts: source, channel and detection, and different encoding or decoding schemes are embodied in these three parts. There are normally two types of channels in practical applications: free space and optical fibers. Theoretically, the safety of the system does not rely on the physical implementation of quantum channels. Technically, metropolitan QKD can take advantage of existing channels and associated infrastructures. Thus, most of the research on integrated QKD focused on sources, modulators, and circuits. Here, we mainly pay attention to the functional devices of source, detection, encoding, and decoding.

### 3.1. Encoding and Decoding

For qubit-based QKDs, quantum information is usually encoded in two orthogonal quantum states and their relative phases—for example, the use of orthogonal polarizations or distinct time bins. Both encoding methods allow robust quantum communication between chips via optical fibers. From an experimental point of view, polarization encoding means modulating the phase of two orthogonal modes of one pulse, whereas time-bin encoding means dividing one pulse into two bits with different time slots and modulating the phases of front and rear bits. The encoding and decoding processes of the polarization scheme and associated devices are relatively simple. However, there will be phase changes between two modes at the receiver due to imperfections of infrastructure like birefringence of fibers. By using time-bin encoding, phases of bits at different time slots are modulated; thus, time delay is needed at the decoder to obtain synchronous phase superposition. This time delay is easily disturbed by environmental factors such as temperature and becomes inaccurate, so the requirements for experimental conditions and devices used in decoding process are relatively higher.

In Table 1, we summarize a list of works of integrated devices for QKD encoding and decoding. An early work of integrated time-bin encoding and decoding implementation [55] was reported by researchers from the Institute of Semiconductors, Chinese Academy of Sciences (IOS-CAS). An asymmetric Mach–Zehnder interferometer (AMZI) was integrated on Si substrate, and its interference visibility was higher than 98%. As the technology developed, more components were fabricated on Si chips, and the interference visibilities of devices were gradually improved [58,63,64,65,74]. A new work reported a decoding chip, which was composed of two AMZIs, three variable optical beam splitters, and four variable directional couplers. The corresponding interference visibility reached 98.6% under temperature control [76]. Long-distance tests with integrated devices for time-bin encryption and decryption were also carried out. For example, time-bin coding devices made of PLCs were used in a QKD field trial, and the quantum bit error rate (QBER) was 2.8% after 97 km transmission [59]. Integrated AMZIs were also used for single-photon interference, and its fringe visibility was more than 80% after 150 km propagation [56]. In addition, a series of integrated components for time-bin encryption or decryption were achieved [57,66,68,69,73,77], including wavelength division multiplexing systems [60,61], matrix switch [67], and polarization-insensitive interferometers [75], etc.

Polarization encoding and decoding devices were also integrated based on different platforms. For instance, a transceiver with a polarization extinction ratio greater than 20 dB was fabricated onto Si substrate [62]. Si-based devices included a dynamic polarization controller, the dynamic polarization extinction ratio of which was greater than 25 dB [79]. Polarization beam splitters based on silica PLC platforms were used for free-space QKD applications [70,71]. Ten Bell-state analyzers were integrated into a FLDW photonic chip for future MDI-QKD applications [72].

For a qudit-based QKD, quantum information is encoded into d (d > 2) orthogonal states. OAM [78], which contains a large Hilbert space, is a typical choice for the integration of qudit-based QKD encoding. To the best of our knowledge, a path-encoding and path-decoding system is also realized for integrated qudit-based QKD [112]. Other coding methods, such as multiple time bin, are also applied to the macroqudit-based QKD and are expected to develop toward integration systems.

### 3.2. Photon Source

In the process of quantum coding, a QKD scheme usually needs to be associated with a quantum source. Here, we mainly outline the quantum photon sources used in QKDs (Table 2) and divide them into three categories: single-photon sources, weak coherent sources, and entangled-photon sources.

Theoretically, each optical pulse emitted by an ideal single-photon source contains only one single photon. But in practical application, it is very difficult to realize devices that can truly generate a single photon per pulse on demand. A promising study was an ultrabright solid-state single-photon source that was reported recently [88]. This photon source was achieved by integrating an atomic defect in hexagonal boron nitride (hBN) associated with a solid immersion lens; it could generate over ten million photons per second at room temperature.

In practical QKD applications, weak coherent sources, which can be easily obtained by attenuating laser emissions, are widely used to approximate single-photon sources. Weak coherent sources were mainly integrated onto InP substrates. One research study showed the visibility of Hong–Ou–Mandel interference between weak coherent states, which were generated by two independent InP transmitters at 431 MHz, was about 46% [80]. A similar result was also observed on two independent III-V lasers at 100 MHz [82]. A recent study [81] reported a Bragg reflection laser that exhibited a minimum inherent linewidth of 10 kHz and an experimental linewidth of 45 kHz by using a delayed self-heterodyne method. The laser operated under single-mode suppression, and the side mode rejection ratio was 54 dB.

Entangled-photon pairs can be generated based on nonlinear optics, i.e., spontaneous parametric downconversion (SPDC) [84] and spontaneous four-wave mixing (SFWM). By using SFWM, photon pairs could be created in microring resonators [83,86] or spiral structures [85,87]. The entangled-photon source (94% visibilities, 3% quantum error rate, 110 bit/s sifted key rate) based on SPDC was integrated into an LN-Si hybrid platform, whereas SFWM-based entangled-photon sources were mainly fabricated on Si substrates. A newer example [85] that demonstrated an Si chip containing more than 550 photonic components and including 16 identical photon-pair sources provided candidates for HD QKDs.

### 3.3. Detection

In the QKD schemes, the receivers use single-photon detectors to read out the quantum information prepared by senders. Single-photon detectors’ technologies involve avalanche photodiodes (APDs) [168,169,170] and superconducting nanowire single-photon detectors (SNSPDs) [171,172,173,174]. As in the case with APDs, traditional SNSPDs are difficult to integrate with other photonic circuits. Hence, the single photons are typically coupled out of the photonic circuit chip into a fiber before being coupled into single-photon detectors. Recently, the development of waveguide-integrated SNSPDs [175] provide a way to integrate full photonic circuity for a QKD receiver. Table 3 lists the QKD systems containing integrated waveguide SNSPDs. In 2011, Si-waveguide SNSPDs were used in the first optimal Bell-state measurement of time bin-encoded qubits produced by two independent lasers [90] and paved the way for QKD networks with untrusted relays. Si_3_N_4_-waveguides SNSPDs were used in a QKD experiment at 2.6 GHz clock rate [89]. The dead time of the SNSPDs was lower than 20 ns, and the dark-count rate was lower than 20 Hz.

## 4. QKD Demonstration

In the section above, we mainly reviewed the functional verification of basic integrated devices for QKD implementation. However, a complete QKD scheme also needs to conduct data acquisition after random modulation, and to estimate the key rate according to the results of data acquisition. Post-processing, such as privacy amplification, error correction, verification, and authentication are also required. In this section, we will review research that completed QKD demonstrations.

### 4.1. Qubit-Based QKD

In Table 4, we summarize works of integrated qubit-based QKDs in chronological order. The first Si photonic integrated circuit transmitter was reported for polarization-encoded QKD [93]. In a 1.3 × 3 mm^2^ die area, a pulse generator, a microring intensity modulator, a variable optical attenuator (VOA), and a polarization controller were integrated together (Figure 1a). The QKD experiment was demonstrated over a 5-km fiber link. Its estimated asymptotic secret key rate (EASKR) reached 0.95 kbps at a 10 MHz clock-rate. This work illustrates the potential of manufacturing low-cost, wafer-scale quantum components on an Si photonic platform. Thereafter, another Si optical integrated system was demonstrated by Sibson et al. [94]. This system consisted of a high-speed transmitter and the corresponding decoder. The fast modulation of quantum states was achieved by combining a thermally tuned phase shifter with a carrier-dispersive phase shifter. Three demonstrations were realized: time bin-encoded BB84 (Figure 1d), polarization-encoded BB84 (1 GHz clock rate, 329 kbps EASKR) (Figure 1c), and pulse modulation for COW QKD (1.72 GHz clock rate, 916 kbps EASKR) (Figure 1b) on an optical fiber network over a 20-km distance. The clock rate of the latter Si integrated system is much higher than that of the former one, whereas more components, i.e., the whole QKD emitter, were fabricated into the former system.

A differential phase shift (DPS) QKD demonstration was carried out with a PLC Mach–Zehnder interferometer. Polarization-insensitive operation at 1 GHz was achieved with a 3076 bits/s key creation rate and a 5.0% QBER [106]. A field test of DPS QKD was performed on a 17.6-km fiber network, and stable operation for six hours was realized with a 120-kbps sifted key rate and a 3.14% average QBER [107]. In a field trial of the Tokyo QKD network, a 2 × 2 AMZI made of PLC was used to help realization of the first secure TV conference over a 45-km distance [91]. The first field test of high-speed polarization-encoded QKD was demonstrated by Bunandar et al. [96], and the secret key rate reached was 1.039 Mbps on 103.6-m deployed fiber links and 157 kbps on 43-km deployed fiber links.

The Si-based integrated chip provided a phase-stable platform for high-speed QKD. Other qubit-based QKD demonstrations using Si-based implementations include works carried out by Huawei [97], the Toshiba Cambridge Institute [98], Sun Yat-sen University [99], the University of Padua [100], IOS-CAS [103,105] and the University of Science and Technology of China [104,105].

An Si-based integrated platform has been used to perform QKD demonstrations, but it usually requires discrete optics, external laser sources, and intensity modulators. A hybrid system with the InP integration platform can meet these requirements. For example, an entirely standalone system used for quantum random number generation and QKD at GHz clock rate was achieved by using a hybrid system [102]. In this system, the high-bandwidth photon sources, electro-absorption modulators and detectors were monolithically integrated on an InP chip as transmitter and a quantum random number generator, whereas the receiver part was fabricated on an Si-based substrate for low propagation loss. Complete electronic control and a post-processing program were also equipped. By using this system, a long-term stable QKD was demonstrated.

Another hybrid system (Figure 2) included a 2 × 6 mm^2^ integrated InP transmitter and a 2 × 32 mm^2^ SiO*_x_*N*_y_* photonic receiver circuit [95]. With the exception of detectors and link controllers, all devices were integrated into chips. Three time bin-modulated QKD demonstrations were realized based on this system: BB84 (560 MHz clock rate, 345 kbps estimated secret key rate), COW (0.86 GHz clock rate, 311 kbps estimated secret key rate) and DPS (1.72 GHz clock rate, 565 kbps estimated secret key rate), by using an attenuation equal to 20-km fiber. The performance of this system could be comparable with the most advanced optical fiber QKD system.

The SiO*_x_*N*_y_* integrated receiver used in the investigations reported by Sibson et al. [94,95] was composed of waveguides that were fabricated by depositing and etching alternating layers of silica and Si_3_N_4_ (Figure 2d). Compared to Si waveguides, SiO*_x_*N*_y_* waveguide structures exhibited high index contrast but low propagation loss, as well as low coupling loss with fibers. Compared to silica PLC [92], SiO*_x_*N*_y_* circuits allow for more complexity. Another SiO*_x_*N*_y_* integrated receiver was used for QKD demonstrations of BB84, DPS, and COW schemes, and their secret key rates were 500 kbps, 400 kbps, and 2500 kbps, respectively, at 2 GHz clock rate [101].

An integrated receiver circuit was created on the Si_3_N_4_ photonic platform [89], which consisted of 325 nm Si_3_N_4_ on 3300 nm silica on an Si substrate. This low-loss chip featured all necessary components, including high-performance single-photon detectors, and it was used for a time-bin BB84 demonstration with a 1500 kbps key rate at 2.6 GHz over a 2.5 dB-loss channel.

### 4.2. MDI-QKD

MDI-QKD is a functional scheme that completely closes all security loopholes in the detection of physical implementations, thereby ensuring the security of communication through untrusted nodes. It has outstanding advantages. For example, MDI-QKD is easy to implement [176]. It also allows cost-effective multiuser-integrated systems, in which users only pay for low-cost transmitter chips while sharing the expensive devices (e.g., SNSPDs) among many users in a public, untrusted relay. Moreover, MDI-QKD supports longer transmission distance compared to traditional QKD systems.

Table 5 lists demonstrations of MDI-QKD based on integrated photonic platforms.

Wei et al. [110] first performed a complete demonstration of 1.25 GHz polarization-encoding MDI-QKD by using an Si-based integrated chip (Figure 3). A high distill finite-key secret rate of 31 bps was achieved over a 36-dB loss channel and a 497-bps key rate was obtained over 140-km commercial fiber spools. To the best of our knowledge, this study was the first GHz MDI-QKD with random modulations and represented a crucial step toward quantum communication with untrusted relays. The security of this GHz system against Trojan horse attacks was verified later, and a 137-bps secret key rate over a 24-dB channel loss was achieved [111].

A proof-of-concept integrated MDI-QKD system was announced by Cao et al. [108]. This system included two Si-based integrated transmitter chips, which operated at 0.5 MHz, and a server chip. Polarization-encoded weak coherent states could be produced with extinction ratios over 20 dB, which promised low-error MDI-QKD. In the proof-of-concept experiment, the system showed a key rate of 1.46 bps over a distance corresponding to 50-km fiber.

The first integrated relay server for MDI-QKD was demonstrated by using a heterogeneous superconducting Si photonic chip [90]. Taking advantage of the high-speed response of superconducting detectors, the optimal Bell-state measurement for time-bin qubits was performed for the first time. Combined with time-multiplexing technology, a secret key rate of 6.166 kbps at 125 MHz was realized over a 24-dB loss channel. This result was comparable to the state-of-the-art MDI-QKD experimental outcomes with a GHz clock rate. Another time bin-encoding MDI-QKD demonstration was performed on a system containing an InP chip, which allowed mass integration of cost-effective devices. Entirely on-chip components were used to prepare high-fidelity 250-MHz clocked weak coherent states. The demonstration showed a 1-kbps estimated secret key rate over a 100-km emulated fiber link, and the QBER was lower than 0.5% [109].

### 4.3. Other Integrated QKDs

There are a few studies related to qudit-based QKD, CV-QKD, and QKD security analyses, so we sorted them into the same table (Table 6).

Because qudit-based QKDs are developed in high-dimensional Hilbert spaces, they are also referred to as high-dimensional (HD) QKDs. Compared to qubit-based QKDs, qudit-based QKDs have better error tolerance and a higher quantum key rate per particle. However, qudit-based QKDs are ordinarily hard to implement in practice due to the difficulty in preparing high-fidelity qudit states. There are two noticeable exceptions called the Chau15 scheme [26] and the round-robin differential phase-shift (RRDPS) scheme [31], which were proposed recently. A qudit-based QKD was proposed and experimentally demonstrated based on space division multiplexing in multi-core fiber by using Si-based integrated path-encoded chips [112]. The use of the multi-core fiber enabled an efficient method for HD quantum state generation at a 5-kHz clock rate. Si-based chips allow stable manipulation of HD quantum states. Three mutually unbiased results were realized in a four-dimensional Hilbert space. A stable quantum bit error rate below coherent attack and individual attack limits was obtained. In an actual decoy, four-dimensional BB84 experiment, the calculated key rate is 7.5 bps over the 4-dB loss channel.

In CV-QKD, the secret keys are encrypted in quadratures of the quantized electromagnetic field and decrypted by coherent detections. Coherent detection is compatible with existing communication infrastructures. Furthermore, it requires no cooling, and its detection efficiency is quite high. Thus, coherent detection becomes a potential application to realize practical quantum cryptography [177]. CV-QKDs can be divided into different classifications according to whether the detection methods are homodyne [178] or heterodyne detections [179], and whether the modulation schemes are Gaussian [180] or discrete modulations [181] etc. An experimental verification of CV-QKD was performed based on an Si photonic chip, in which all necessary components were integrated except the photon source [113]. This chip implemented Gaussian-modulated coherent state and was used for CV-QKD demonstration over a 2-m fiber. The resulting key rate was 0.25 Mbps. The feasibility of long-distance CV-QKD was also simulatively verified.

To date, only some theoretical models of QKD systems based on specific assumptions can be proven to have unconditional security. In fact, there are always various threats in the practical applications due to imperfections in the source [182,183,184], modulation [185,186], and detection [187,188,189,190]. In order to apply the unconditional security proof of the theoretical model to the actual QKD systems, it is necessary to build the theoretical model based on assumptions that can be verified by experiments and further develop corresponding tests meant to verify these assumptions—that is, security analyses, which include security definitions, implementation assumptions, and diverse security proofs. These issues are well studied for a bulk optic system [191,192,193,194,195,196,197]. The security analyses are also crucial for the chip-based QKD at a starting state. Here, we review some security analyses performed on integrated photonic QKD systems.

Although many QKD schemes have been demonstrated on integrated photonic chips, polarization-dependent loss in state preparation has not been considered in the key-rate estimation process. One recent study [116] illustrated that a large amount of polarization-dependent loss existed in realistic Si-based integrated state-preparation devices and might compromise system security. Then a decoy-state BB84 QKD experiment that took polarization-dependent loss into account was carried out, and a rigorous finite-key security bound was obtained over a fiber link up to 75 km. A security analysis of influence caused by carrier fluctuations combined the plasma dispersion effect and was performed on an Si-based integrated modulator used for CV-QKD [114]. Two preliminary defense strategies were proposed: the maximum carrier fluction deviation method and dynamic random carrier fluction calibration based on the deep neural network. The finite-key security of BB84 and MDI-QKD was analyzed based on an Si-integrated chip with information leakage out of polarization modulators and intensity modulators. With a 232-dB isolation, the MDI-QKD system was still able to resist the Trojan horse attack [115].

## 5. Conclusions and Outlook

In this review, we first introduced a basic QKD background (Section 1) and the development of integrated platforms applied to quantum photonics (Section 2). Then the performance of integrated quantum devices (Section 3) and a series of experimental QKD verifications based on photonic integrated chips (Section 4) were discussed.

Combined with various integration technologies, QKD systems will scale up quickly and promise more complexity and functionality. Here, we provide several suggestions for the focus of further researches.

Security analyses: The unconditional security of QKD is based on the hypothesis that the practical system model is consistent with the theoretical model used in the security proof. In fact, there are ineluctable differences between theoretical and practical system models that lead to security vulnerabilities in the QKD system. See reference [198] for a short overview of this topic. For integrated photonic QKD systems, further security analyses, including calibrations or tests of whether the chips meet the safety assumptions and designs of secure testable systems, are still necessary.Higher-level integrations of quantum photonic systems: We have described some QKD photonic chips in this review, and one can see that there is not a complete integrated system. In the future, complete integrated systems should be realized with the development of hybrid integration technologies.Demonstrations of advanced QKD protocols: Previous studies of QKD demonstrations with integrated photonic chips were mostly based on classic protocols like BB84 or MDI. In the future, we should pay attention to combinations of integration technologies and cutting-edge QKD protocols, such as the recently proposed twin-field QKD [199,200], which can break through the PLOB bound [201], as well as its derivative protocols like phase-matching QKD [202], sending-or-not-sending [203], and no-phase-postelection protocols [204].

## Figures and Tables

**Figure 1 entropy-24-01334-f001:**
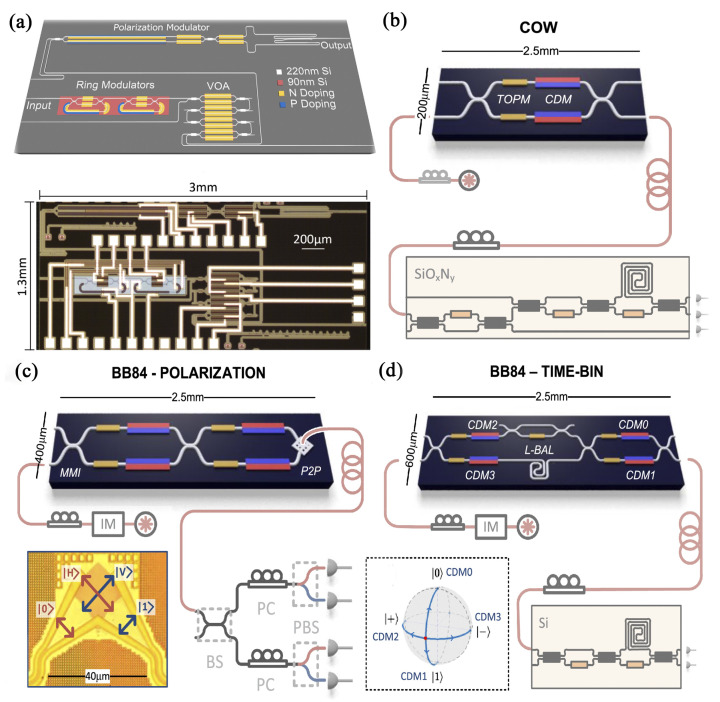
Integrated photonic chips for qubit-based QKDs. (**a**) Schematic and optical micrograph of the first Si-based integrated transmitter for polarization-encoded QKD. In a 1.3 mm × 3 mm die area, a pulse generator, a microring intensity modulator, a VOA, and a polarization controller were integrated together. (**b**) Chip-to-chip QKD for realization of COW. Beam splitters, thermo-optic phase modulators (TOPMs) and carrier-depletion modulators (CDMs) are integrated in an Si chip, which is used for the encoding of quantum information in path, or pulse modulation. (**c**) Polarization-encoded BB84. Two multimode interferences (MMIs) act as the two paths of an MZI, combining with a 2D grating coupler, for the conversion from path-encoded information to polarization-encoded information (P2P). (**d**) Time bin-encoded BB84. Four CDMs used for fast modulation. L-BAL stands for loss-balancing. Panels reproduced from: (**a**) [93]; (**d**) [94] under a Creative Commons licence (https://creativecommons.org/licenses/by/4.0/accessed date: 29 July 2022).

**Figure 2 entropy-24-01334-f002:**
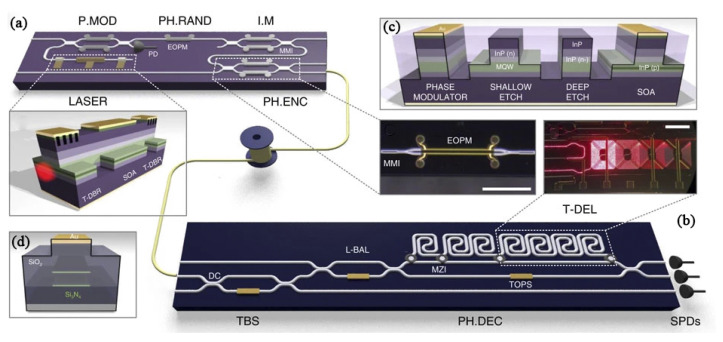
A hybrid system including an integrated InP transmitter and an SiO*_x_*N*_y_* receiver. The system is used for reconfigurable, multi-protocol QKD. (**a**) The InP transmitter chip. (**b**) The SiO*_x_*N*_y_* receiver circuit. (**c**) Cross-section of the InP deep-etch waveguides. (**d**) Cross-section of the SiO*_x_*N*_y_* Triplex waveguides. Panels reproduced from [95] under a Creative Commons licence (https://creativecommons.org/licenses/by/4.0/ accessed date: 29 July 2022).

**Figure 3 entropy-24-01334-f003:**
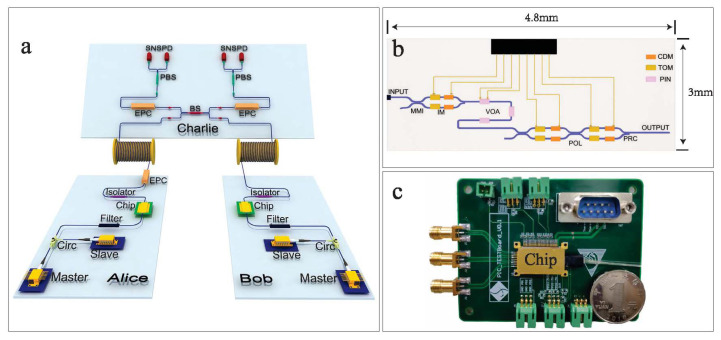
An example of Si-chip-based photonic MDI-QKD system. (**a**) Experimental setup. (**b**) The schematic of the Si chip, in which multimode interference (MMI) couplers, thermo-optics modulators (TOMs), carrier-depletion modulators (CDMs), and a polarization rotator combiner (PRC) are integrated. (**c**) Image of the packaged chip soldered to a compact control board. Panels reused from Ref. [110].

## Data Availability

Not applicable.

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
