# Peer review of "Advances in Chip-Based Quantum Key Distribution"

_entropy, 2022, doi:10.3390/e24101334_

Round 1

Reviewer 1 Report

In this review paper entitled “Advances in Chip-based Quantum Key Distribution”, the authors introduced and summarized the recent advances in integrated quantum key distribution systems, including the development of integrated platforms applied to quantum photonics, the performance of integrated quantum devices and a series of experimental QKD verification based on photonic integrated chips

 Recently, the main progress of quantum key distribution is focused on the on-chip devices of QKD. And it is becoming quite an important field of quantum communications which attracts much attention during the past decades. I think the current manuscript is interesting and timely which may provides the readers detailed materials of the research field in chip-based QKD. I suggest that the current manuscript could be accepted after minor revision. Here in the following, I list some related issues that needs to be considered:

The security is also the main issue in chip-based QKD, could you give a short introduction and discussion about the progress on that?

Author Response

A short introduction about security analyses on chip-based QKD were added in Section 4.3 as the 4th paragraph. The new text reads:

“To date, only some theoretical models of QKD systems based on specific assumptions can be proved to have unconditional security. But in fact, there are always various threats in the practical applications due to the imperfections in source [182–184], modulation [185,186], and detection [187–190]. In order to apply the unconditional security proof of the theoretical model to the actual QKD systems, it is necessary to build the theoretical model based on assumptions that can be verified by experiments, and further develop corresponding tests mean to verify these assumptions. That is, security analyses, which include the security definitions, implementation assumptions and diverse security proofs. These issues are well studied for a bulk optic system [191–197]. The security analyses are also crucial for the chip-based QKD and it is at a starting state. Here, we review some security analyses performed on integrated photonic QKD systems.”

Reviewer 2 Report

QKD can share the random key between two uses. It provides us the approach to realize the secure communication. QKD was widely realized in space, fiber. The Chip-based QKD was also well developed. In this review, the authors provide a complete review about Chip-based QKD. This manuscript is well written and can be published after some minor revisions.

1 The abstract is not well written. “we summarize the advances in integrated quantum key distribution systems”. It is better to add some explain about the integrated quantum key distribution systems in this review.

2 In the introduction, the authors claim that “there are two methods to quantum secure encryption scheme.” Actually, besides the PQC and QKD, there is another important secure communication approach, named QSDC. QSDC can transmit the secure message without sharing a public key. Certainly, it can also distribute the key, acts as the same role as QKD. I suggest the authors add some introduction about QSDC in this review.

3 There are some work about QKD are suggested.

1L. C. Kwek, L. Cao, W. Luo, Y. Wang, S. Sun, X. Wang, and A. Q. Liu, AAPPS Bull. 31, 15 (2021).

2 Z. Q. Yin, F. Y. Lu, J. Teng, S. Wang, W. Chen, G. C. Guo, and Z. F. Han, Fundam. Res. 1, 93 (2021).

3 H. Guo, Z. Li, S. Yu, and Y. Zhang, Fundam. Res. 1, 96 (2021).

4 G. Z. Tang, C. Y. Li, and M. Wang, Quant. Eng. 3, e79 (2021).

5 X. Wang, X. Sun, Y. Liu, W. Wang, B. Kan, P. Dong, and L. Zhao, Quant. Eng. 3, e73 (2021).

Author Response

  1. Some details about integrated quantum photonics and integrated QKD systems were added into the abstract. The abstract was revised as follow:

“Quantum key distribution (QKD), guaranteed by the principle of quantum mechanics, is one of the most promising solutions for the future secure communication. Integrated quantum photonics provides a stable, compact and robust platform to implement complex photonic circuits amenable to mass-manufacture, while also allows the generation, detection and processing of quantum states of light at a growing systems’ scale, functionality and complexity. Integrated quantum photonics provides a compelling technology for integration of QKD systems. In this review, we summarize the advances in integrated QKD systems, including integrated photon sources, detectors, encoding and decoding components for QKD implements. Complete demonstrations of various QKD schemes based on integrated photonic chips are also discussed.”

  1. A short introduction about QSDC was added into the Section 1.1. The new text reads:

“The second method is quantum secure direct communication (QSDC), which was first proposed by Long and Liu in 2002 [10] based on quantum laws. QSDC allows users to directly transmit private information over secure quantum channels without security key distribution. Due to its relatively simple communication steps, QSDC has drawn great attention and has rapid developments [11–14] in the past two decades. However, the achieved key rates of QSDC [15–18] are rather low than other quantum secure encryption schemes.”

  1. Suggested paper 1 was cited as Ref. [165];

Suggested paper 2 was cited as Ref. [200];

Suggested paper 3 was cited as Ref. [177];

Suggested paper 4 was cited as Ref. [176];

Suggested paper 5 was cited as Ref. [18];

Reviewer 3 Report

The manuscript introduces recent advances of integrated quantum key distribution, which provides a valuable reference for the further work. However, there are some problems to be solved. I would recommend it for publication, conditioned on revising the following issues that I raise.

1. The abstract is an important part of the manuscript, which should show the main result and advantages of the manuscript. The abstract of the manuscript is very poor. The contents of the abstract about integrated photonics and QKD is not detailed enough. The authors should enrich it. 

2. Many paragraphs are over described. For example, the subsection 1.2 and the last two paragraphs of the subsection 1.1 have little correlation with integrated photonics and QKD. The authors should keep the references but make contents more brief.

In subsection 4.3, the contents are divided into too many paragraphs. The authors should also make them more brief.

3. The authors leave out some references about single photon detectors in subsection 3.3. I have only seen two references about SNSPDs. To increase the readability of the manuscript, the authors need to extensively investigate more about this part.

Author Response

  1. Some details about integrated quantum photonics and integrated QKD systems were added into the abstract. the new abstract was listed above that won’t be repeated here.
  2. The last two paragraphs of the subsection 1.1 give a brief overview about current quantum secure encryption schemes: PQC and QKD, and serve as a link between the preceding and the following. Moreover, according to the second comment of reviewer 2, we believe that reviewer 2 also consider these two paragraphs are necessary and even suggest to add a paragraph about QSDC. Therefore, we insist on retaining these two paragraphs.

Subsection 1.2 give a short introduction about QKD and all the QKD protocols that will be discussed in the following sections. The main differences between various QKD protocols are briefly explained. Therefore, we believe that subsection 1.2 is necessary to retain.

The paragraphs in subsection 4.3 have been simplified and merged according to different QKD schemes.

  1. In quantum optics, single photon detectors are difficult to integrate and expensive. Relevant research is still in its infancy. Therefore, there are only two papers about integrated single photon detectors in a strict sense as we described in the first draft. According to the comments of the reviewer 3, to increase the readability of the manuscript, we briefly expanded the discussion about APDs and SNSPDs, relative references were also added. Text for substitution reads:

“In the QKD schemes, the receivers use single photon detectors to read out the quantum information prepared by senders. Single photon detectors’ technologies involving avalanche photodiodes (APDs) [168–170] and superconducting nanowire single photon detectors (SNSPDs) [171–174]. As in the case with APDs, traditional SNSPDs are difficult to integrate with other photonic circuits. Hence the single photons are typically coupled out of the photonic circuit chip into a fiber before being coupled into single photon detectors. Recently, the development of waveguide-integrated SNSPDs [175] provide a way to integrate full photonic circuity for QKD receiver. Table 3 lists the QKD systems containing integrated waveguide SNSPDs. In 2011, Si-waveguide SNSPDs were used in the first optimal Bell-state measurement of time-bin encoded qubits produced by two independent lasers [90], paved the way for QKD networks with untrusted relays. Si3N4-waveguides SNSPDs were used in a QKD experiment at 2.6 GHz clock rate [89]. The dead time of the SNSPDs was lower than 20 ns and the dark-count rate was lower than 20 Hz.”
